

# Multi-year observations of variable incomplete combustion in the New York megacity

Luke D. Schiferl[1], Cong Cao[2], Bronte Dalton[3,4], Andrew Hallward-Driemeier[1,5], Ricardo Toledo-Crow[6], and Róisín Commane[1,5]

[1]Lamont-Doherty Earth Observatory, Columbia University, Palisades, NY 10964, USA
[2]School of Marine and Atmospheric Sciences, Stony Brook University, Stony Brook, NY 11794, USA
[3]Columbia College, Columbia University, New York, NY 10027, USA
[4]Institute for Health Metrics and Evaluation, University of Washington, Seattle, WA 98195, USA
[5]Department of Earth and Environmental Sciences, Columbia University, New York, NY 10027, USA
[6]Advanced Science Research Center, City University of New York, New York, NY 10031, USA

*Correspondence to*: Luke D. Schiferl (schiferl@ldeo.columbia.edu)

**Abstract.** Carbon monoxide (CO) is a regulated air pollutant that impacts tropospheric chemistry and is an important indicator of the incomplete combustion of carbon-based fuels. In this study, we used four years (2019–2022) of winter and spring (January–May) atmospheric CO observations to quantify and characterize city-scale CO enhancements ($\Delta$CO) from the New York City metropolitan area (NYCMA). We observed large variability in $\Delta$CO, roughly 60% of which was explained by atmospheric transport from the surrounding surface areas to the measurement sites, with the remaining 40% due to changes in emissions on sub-monthly timescales. We evaluated the CO emissions from the Emissions Database for Global Atmospheric Research (EDGAR), which has been used to scale greenhouse gas emissions, and found the emissions are much too low in magnitude. During the COVID-19 shutdown in spring 2020, we observed a flattening of the diurnal pattern of CO emissions, consistent with reductions in daytime transportation. Our results highlight the role of meteorology in driving the variability of air pollutants and show that the transportation sector is unlikely to account for the non-shutdown observed CO emissions magnitude and variability, an important distinction to determine the sources of combustion emissions in urban regions like the NYCMA.

## 1 Introduction

Carbon monoxide (CO) is released into the atmosphere from incomplete combustion of carbon-based fuels. Poisonous to humans at high concentrations, CO is a regulated criteria pollutant in many countries including the United States (US). CO emissions come from both anthropogenic (e.g., transportation, manufacturing, power generation) and natural (e.g., wildfires, biomass burning) combustion sources. CO emissions from on-road vehicles in the US have decreased by over 50% in the past several decades due to improvements in combustion efficiency (Parrish, 2006; Bishop and Stedman, 2008). These emissions reductions have led to lower urban CO concentrations measured in situ at the surface and by satellites throughout the column (He et al., 2013; Pommier et al., 2013; Yin et al., 2015; Hassler et al., 2016; Hedelius et al., 2021; Buchholz et al., 2021). US



cities rarely experience highly toxic concentrations of CO in recent years, but CO is still a useful tracer for incomplete combustion (e.g., Turnbull et al., 2015). Recent improvements in efficiency may be slowing, however, perhaps due to diminishing returns in catalytic converters (McDonald et al., 2013; Jiang et al., 2018).


CO also impacts atmospheric chemistry by controlling oxidative capacity (OH), which in turn impacts the lifetime of methane ($CH_4$) and ozone ($O_3$) production. CO is therefore also an indirect greenhouse gas. The decreasing trend in the global CO burden has led to a relative increase in CO chemical production, higher methane oxidation by OH, and a shorter $CH_4$ lifetime (Gaubert et al., 2017).


Given the observed changes to atmospheric CO and their potential implications for chemistry and climate, it is important to consider the timescales over which urban CO emissions are changing and which factors control their variability. The recent decreasing annual trends in CO emissions, such as ~4.6%/year for the US during 2002–2011 (Yin et al., 2015) and ~4.5%/year for the Washington DC/Baltimore region during 2015–2020 (Lopez-Coto et al., 2022), have been attributed to improving

vehicle combustion efficiency. Over multiple years of flights Lopez-Coto et al. (2022) also observed consistently lower CO emissions on Sundays, likely from fewer on-road vehicles on these days. On shorter scales, Ren et al. (2018) found significant variability (+/– 50–70% of the mean) in CO emission rates between flights for winters 2015 and 2016, but did not explain the variability. Lopez-Coto et al. (2020) attributed observed variability between several days of flights to the varied sampling of sources that also have high hourly variability, such as power plants and traffic. Hall et al. (2020) found evidence for a vehicle

emissions dependence on meteorology, where CO emissions increased with increasing temperature and specific humidity across the winter-to-summer gradient in the DC/Baltimore region.

CO inventories track the expected emissions changes over time and are frequently used in two ways, despite their uncertainty and limited temporal resolution: (1) as an *a priori* emission for geostatistical and Bayesian inversion studies which calculate

optimal emissions relative to the CO inventory (thereby providing an indirect evaluation of the inventory) and (2) to scale greenhouse gas emissions relative to observed atmospheric ratios. The US Environmental Protection Agency (EPA) National Emissions Inventory (NEI) is often used in inversion studies, is updated every few years for reactive compounds including CO, and has been evaluated substantially: NEI CO emissions are overestimated up to 2 times compared to observations over multiple inventory years (Brioude et al., 2013; Miller et al., 2008; Salmon et al., 2018; McDonald et al., 2018), although some

have found smaller discrepancies (Anderson et al., 2014; Castellanos et al., 2011; Gonzalez et al., 2021; Lopez-Coto et al., 2022; Brioude et al., 2011). The Emissions Database for Global Atmospheric Research (EDGAR), which provides monthly global CO, $CH_4$, and $CO_2$ emissions, is less often evaluated, despite being used extensively for scaling greenhouse gas emissions. Greenhouse gas scaling methods use the observed atmospheric ratio of $CO_2$:CO or $CH_4$:CO along with the inventory CO emission rate to calculate an observation-derived $CO_2$ or $CH_4$ emission rate (Wunch et al., 2009; Hsu et al., 2010).

Konovalov et al. (2016) used CO satellite column measurements to derive scaled $CO_2$ emissions based on the $CO_2$:CO from



EDGAR v4.2 (and others). Recently, CO emissions from EDGAR were also used to scale CH$_4$ emissions from cities across the US based on observed CH4:CO ratios from in-situ aircraft measurements (v4.3.2, Plant et al., 2019; Ren et al., 2018) and satellite columns (v5.0, Plant et al., 2022). These works generally assume that CO emissions are less uncertain than CH$_4$ emissions. In one of the few evaluations of EDGAR CO emissions, Ren et al. (2018) found the inventory rate to be within the

uncertainty of their mass balance approach. CO emission inventories with higher spatial and temporal resolution than EDGAR and NEI are also available, but they are regional and sector specific (e.g., for mobile emissions (McDonald et al., 2014; Gately et al., 2017)) and are therefore more difficult to evaluate when mixed with other emission sectors in the atmosphere. Long term in-situ CO observations provide an advantage for evaluating seasonal trends in inventories compared to aircraft snapshots and satellite observations but have so far been underutilized.


Restrictions on movement imposed by the COVID-19 pandemic beginning in March 2020 greatly reduced transportation-related emissions from both commuting and leisure activities. These reductions were noted on the global scale for CO$_2$ (Le Quéré et al., 2020; Forster et al., 2020; Liu et al., 2020) and on the city scale for NO$_2$ (Goldberg et al., 2020; Tzortziou et al., 2022; Shi et al., 2021), CO (Lopez-Coto et al., 2022), and CO$_2$ (Turner et al., 2020). City centers also experienced an overall

reduction in human activity (with potential implications for building energy and electricity consumption) as suburbanites worked from home during the day and many city-dwellers fled to rural areas, but the impact of these population pattern changes on combustion emissions is less clear. The New York City metropolitan area (NYCMA) was the first region in the US impacted by COVID-19 shutdowns. Mobility data indicated a drop of up to 60% in traffic and 90% in public transit metrics between February and April 2020 (Forster et al., 2020; Tzortziou et al., 2022; Cao et al., 2023). Tzortziou et al. (2022) examined NO$_2$

changes in the NYCMA and found an average reduction of 32% in the city center during the peak shutdown period (15 March–15 May 2020). Concentrations of VOCs from both combustion (e.g., benzene) and non-combustion (e.g., personal care products, industrial chemicals, oxidation products) sources also declined by similar magnitudes during this time (Cao et al., 2023). Residual shutdowns (e.g., school closures) stayed in place until March 2021, with mobility, total column NO$_2$, and many VOC observations remaining lower than pre-shutdown.


In this study, we analyze multi-year atmospheric CO observations at two sites in the urban core to quantify and characterize the variability of city-scale CO enhancements (ΔCO) from the NYCMA over sub-monthly and sub-daily timescales. We use the observed ΔCO along with an atmospheric transport model to isolate the impacts of meteorology on the observations and evaluate the EDGAR inventory, which is widely used to scale greenhouse gas emissions. We also identify changes to regional

CO emissions induced by the peak and residual COVID-19 shutdowns. Using CO as a tracer, this work begins to constrain the uncertainty in urban combustion sources.



## 2 Methods

### 2.1 Rooftop Observations

Ambient CO dry-mole fractions (units: ppbv, parts-per-billion by volume) were measured at the Advanced Science Research
Center (ASRC) Rooftop Observatory, a site located 56 m above ground level (93 m above sea level (ASL)) in Hamilton
Heights, West Harlem, Manhattan (40.81534°N, 73.95033°W) (Fig. S1). The site samples a mixture of combustion sources
including on- and off-road transportation, building energy, manufacturing, and electricity generation. Additional description
of the ASRC site is found in Commane et al. (2023), and other in situ observations from this site were used by Cao et al.
(2023).


Due to varying availability, several different instruments were used to measure CO over the four subsequent winters and
springs (January–May) of the study period (2019–2022). The instruments used in this study are: (i) Picarro G2401-m for 2019
and 2020 (reporting at 0.5–1 Hz), (ii) Picarro G2401 for 2021 (reporting at ~0.3 Hz), and (iii) Aerodyne SuperDUAL for 2022
(reporting at 1 Hz). Each instrument was calibrated using gas cylinders that are traceable to standards calibrated by the Central
Calibration Laboratory (CCL) at the National Oceanographic and Atmospheric Administration (NOAA) Global Monitoring
Laboratory (GML) in Boulder, CO. CCL maintains the World Meteorological Organization (WMO) CO scale (WMO
X2014A). The Aerodyne SuperDUAL set-up at ASRC is described by Commane et al. (2023).

We calculate the hourly mean of these CO measurements at the ASRC site for hours with at least 50% valid sub-hourly
measurements (e.g., at least 1800 1-Hz measurements).

### 2.2 Network Observations

We also use hourly CO dry-mole fractions at two EPA Air Quality System (AQS) network sites: (i) the City College of New
York (CCNY, a campus of the City University of New York or CUNY) (40.81976°N, 73.94825°W) located in Manhattan 500
m north of the ASRC site and (ii) Cornwall (41.82134°N, 73.29726°W) located in Connecticut ~125 km northeast of the other
sites (Fig. S1). The CCNY site uses a Teledyne API 300EU analyzer, which has a detection limit of 20 ppb and is calibrated
weekly. The Cornwall site uses a Thermo Scientific 48i-TLE analyzer, which has a detection limit of 40 ppb and is auto-
calibrated daily. EPA CO observations are calibrated to the National Institute of Standards and Technology (NIST) scale,
which is within 5 ppb of the NOAA/WMO calibration scale at ambient dry-mole fractions (Lee et al., 2017). Neither time-
specific uncertainties nor sub-hourly observational variability were reported at either site. The CCNY site is located 45 m ASL
and is classified by the EPA AQS as an urban-scale site measuring within 4-50 km, while the Cornwall site is located 505 m
ASL and is a regional-scale site sampling air within 50–100s km of the site.



**2.3 Isolation of City-scale Measurements**

We use the proximity of the ASRC and CCNY sites (0.5 km apart) to separate the CO measurements representative of the city scale, which we are interested in for characterizing the CO variability of the NYCMA, from the local scale, emissions from

nearby sources like buildings or roads. Based on the assumption that city-scale observations should be similar at both sites, we define city-scale measurements at each site as any hourly CO observation less than or equal to 3 times the standard deviation ($\sigma$) added to the daily mean of the other site. In contrast, local-scale observations have variability greater than the mean $+ 3\sigma$ of the other site. In this categorization, we only use hours when both ASRC and CCNY provide CO observations.

The hourly observed CO mole fractions are highly variable and range from 100 ppb to 3000 ppb at the ASRC site, while the CCNY site observes fewer high peaks (maximum 2000 ppb) (Fig. 1). Our categorization scheme indicates that many of these observed peaks are from local sources nearby to the measurement sites, rather than representative of the broader city scale. As with the observed peaks, these local-scale measurements are more prevalent at ASRC than at CCNY, especially during 2020–2022. A comparison of the city-scale CO observations shows good correlation ($R^2$=0.61, slope=0.82) between the sites,

consistent with using the $3\sigma$ filter (Fig. S2). The local-scale observations are clearly outliers in this relationship, especially the high CO mole fractions at ASRC. We exclude the local-scale observations from the subsequent city-scale analysis. This simple filtering approach is conservative and retains ~80% of the hourly CO measurements at city scale.

**2.4 Background Estimation**

To characterize and quantify CO from the NYCMA, we must account for the atmospheric CO entering the domain without

impact from the study area ("the background"). Following methods developed in Ammoura et al. (2016) and Lopez-Coto et al. (2022), we estimate the hourly background CO at the ASRC and CCNY sites as the fifth percentile of mole fractions from the previous and following five days (ten days total) using only the city-scale CO observations. The background CO mole fractions are only calculated for hours with at least 50% valid CO data in the ten-day window (i.e., at least 120 valid hours).

Given its location on the edge of the NYCMA domain and status as a regional measurement site, we use the hourly CO mole fractions from the Cornwall site as an independent estimation of the background CO into the NYCMA domain. The hourly background CO at Cornwall is defined as the mean of the CO observations from the previous and following five days (ten days total). As done for the ASRC and CCNY site backgrounds, the Cornwall background CO mole fractions are only calculated for those hours with at least 50% valid CO data in the ten-day window.


The background CO mole fractions are variable but tend to peak in late winter (Fig. S3). Differences in the background CO calculated from the various sites for a given time (i.e., the uncertainty) may be up to 50 ppb but are often much lower (~5–10 ppb). Some CO emitted from the NYCMA may be sampled at Cornwall on days with strong southwest winds, but the 125 km



distance allows for dilution of any plumes. Using Cornwall as a background may lead to a slight underestimate in the magnitude

of the observed CO enhancements (Sect. 2.5).

**Figure 1.** Time series of hourly CO observations used in the analysis categorized as city-scale (black) and local-scale (red) at the ASRC (left) and CCNY (right) sites for January–June 2019–2022 (top to bottom).



## 2.5 Calculation of Observed CO Enhancements

We define the observed CO enhancement (ΔCO) generated by the NYCMA for each hourly city-scale observation as in Eq. 1:

$$\text{observed } \Delta CO = \text{observed CO} - \text{background CO} \tag{1}$$

where the observed ΔCO (units: ppb) is the observed CO dry-mole fraction with the background CO removed. Observed ΔCO is calculated for each of the ASRC and CCNY sites using the observed CO from those sites and both the fifth percentile background for that site and the Cornwall background. We only calculate observed ΔCO for hours with valid background CO from both locations to quantify the background uncertainty.

From the hourly observed ΔCO, we calculate: (i) the 10-day mean observed ΔCO centered on each day of the study period to assess sub-monthly CO variability while removing variability on synoptic timescales and (ii) the mean observed ΔCO for each two-hour period throughout the day (a diurnal pattern) over the weekdays (Monday–Friday) and weekends (Saturday–Sunday) of various periods of at least 30 days to assess sub-daily CO variability. In each case, the mean observed ΔCO is only calculated for periods with at least 50% valid hourly observed ΔCO.

## 2.6 Emissions Inventory

We use anthropogenic CO emissions from v6.1 of the Emissions Database for Global Atmospheric Research (EDGAR) inventory for air pollutants, which are available monthly for 2018 at 0.1°x0.1° spatial resolution (Crippa et al., 2018, 2020). EDGAR CO emissions are greatest in the center of the NYCMA (NYC in Fig. S1), where the combustion from the building energy (12% of January–May total), road transportation (43%), manufacturing (16%), and power generation (11%) sectors combine to produce substantial CO emissions for the region. Transportation emissions follow along the road network outward to the rest of the domain. Monthly totals for the study domain show peak emissions in February dropping to lowest emissions in May, with April close to the annual mean (Table S1). These month-to-month CO emissions changes are mostly due to the seasonal pattern in building energy usage (i.e., heating). Monthly variability in the EDGAR CO emissions is also spatially explicit, with a larger absolute change month-to-month in the NYC core region of the domain, where the absolute emissions are also largest (Fig. S4). However, the outer areas of the domain with lower emissions totals experience a greater relative change since a larger portion of the CO emissions there are from building energy.

Compared to previous versions, EDGAR v6.1 extends activity data through 2018 and updates emissions factors for combustion and evaporative sources due to road transportation technology improvement. In the NYCMA, total CO emissions in EDGAR v6.1 decline by 25% from 2012 to 2018. EDGAR v6.1 CO emissions are also 26% lower in our domain compared to EDGAR v5.0 for 2015, the latest year available for that version. We do not apply any interannual emissions scaling to the EDGAR





inventory for our study period due to the high uncertainty of sub-national variability, especially during the COVID-19 shutdown periods, and due to inconsistencies between reported trends in EDGAR and NEI (Plant et al., 2022). EDGAR provides emissions uncertainty on a national basis, which most recently is cited as 44% for US CO emissions in 2012 from 200 EDGAR v4.3.2 (Crippa et al., 2018). While Crippa et al. (2020) suggest methods to implement diurnal variability in EDGAR using nationwide sector-specific scale factors, we choose to leave emissions constant throughout the day. Instead, we use the observations to define a city-specific diurnal emissions scaling (see Sect. 2.9).

CO emissions from biomass burning sources are very limited during the winter and spring in the NYCMA, and we do not 205 consider them here. For example, total CO emissions for January–May (for each year in 2012–2018) from the Global Fire Emissions Database v4 with small fires (GFED4s, van der Werf et al., 2017) are less than 1% of the total EDGAR CO emissions during these months.

## 2.7 Transport Model

The Stochastic Time-Inverted Lagrangian Transport (STILT) model simulates the impact surface fluxes have on the 210 atmospheric mole fraction at a given time and place (Fasoli et al., 2018; Lin et al., 2003). STILT moves idealized particles backward in time with their 3-dimensional movement determined by large-scale winds and random turbulence. Contribution of the surface flux to the atmospheric mole fraction (the surface influence) occurs when any particle resides within the lower half of the planetary boundary layer. The surface influence accumulated by each particle is interpolated to a regular grid based on their locations in time to derive a "surface influence footprint".


We drive STILT using NOAA High-Resolution Rapid Refresh (HRRR) meteorology (3 km horizontal, hourly temporal resolution) (Benjamin et al., 2016) and refer to the two components together as HRRR-STILT. We configure HRRR-STILT to calculate the surface influence footprint (0.01° horizontal, hourly temporal resolution) for the NYCMA domain (Fig. S1), initiated for each hour of the study period, by running 500 particles backward for 24 hours from the ASRC observation site. 220 Using these surface influence footprints, we also calculate the relative NYC surface influence (unitless) by normalizing the summed 24-hour footprints for the NYC subdomain.

STILT was previously used by Miller et al. (2008) to assess CO emissions across North America. More recently, STILT has been widely used in anthropogenic greenhouse gas studies in various regions (e.g., Floerchinger et al., 2021; Turner et al., 225 2016; Lin et al., 2021; Sargent et al., 2018), and the HRRR-STILT coupling has been used in several urban areas (e.g., Turner et al., 2020; Ware et al., 2019). Footprints from the HRRR-STILT setup described here were previously used by Tzortziou et al. (2022), Tao et al. (2022), Wei et al. (2022), and Cao et al. (2023) to investigate $NO_2$, ozone, $CO_2$, and VOCs, respectively, in the NYCMA.



**2.8 Calculation of Simulated CO Enhancements**

We determine the simulated CO enhancement (ΔCO) generated by the NYCMA for each hour of the study period as in Eq. 2:

$$\text{simulated } \Delta CO = \text{inventory CO emissions flux} \times \text{surface influence footprint} \tag{2}$$

where the simulated ΔCO (units: ppb) is the EDGAR inventory CO emissions flux (units: nmol m$^{-2}$ s$^{-1}$) multiplied by the 24-
hour HRRR-STILT surface influence footprint covering the NYCMA (units: ppb (nmol m$^{-2}$ s$^{-1}$)$^{-1}$). Given the lifetime of CO
in the atmosphere (~1 month), very limited chemical loss is expected over this 24-hour period, so all surface CO emissions
intercepted by the footprint will reach the observation site.

Each hour produces a single simulated ΔCO, which we sample to match the valid hourly observed ΔCO for each observation
site and background combination. Mean simulated ΔCO is also calculated from the hourly simulated ΔCO as described above
for the mean observed ΔCO. To assess the sensitivity of our methods, addressed below, we retain the unsampled simulated
ΔCO and also calculate hourly and mean simulated ΔCO using the annual mean EDGAR CO emissions. We also test the
sensitivity of simulated ΔCO to the STILT configuration settings impacting mixing (horizontal turbulence, minimum mixing
height, and vertical mixing scheme) and to the choice of meteorological product (NAMS (North American Mesoscale Forecast
System at 12 km horizontal resolution), GFS (Global Forecast System at 0.25°), and GDAS (Global Data Assimilation System
at 1°) for a subset of the study period (January–February 2022) that shows high variability in observed ΔCO. The impact of
mixing scheme and meteorological product selection were previously assessed in GHG studies by Sargent et al. (2021), Hajny
et al. (2022), and Tomlin et al., (2023).

**2.9 Observed Relative Emissions (ORE)**

Given the lack of sub-monthly and sub-daily variability in the inventory CO emissions from EDGAR, any variability in the
mean simulated ΔCO is attributed to variability in the surface influence footprint (i.e., transport meteorology). The remaining
variability in mean observed ΔCO that is not captured by the simulated variability is likely then to be due to changes in the
CO emissions not included in the inventory. We normalize the observed ΔCO by the simulated ΔCO to calculate the observed
relative emissions (ORE, unitless) as in Eq. 3, which removes the impact of meteorology and isolates the observed change in
CO emissions for any period:

$$\text{ORE} = \frac{\text{observed } \Delta CO}{\text{simulated } \Delta CO} \tag{3}$$

The ORE also represents the relative bias in the CO inventory compared to the CO observations, where ORE greater than 1
indicates the CO inventory needs to be increased to match the observations, while ORE less than 1 indicates the inventory is



biased high. Sensitivity experiments (not shown) indicate that on average emissions changes suggested by the ORE reach the ASRC site via transport within 2 hours, so CO emissions and enhancements can be considered to occur simultaneously even at the shortest diurnal timescale examined here.

## 3 Results and Discussion

We use our observations to quantify the magnitude and variability of the city-scale observed CO enhancements (ΔCO) from the NYC metropolitan area (NYCMA). Then, we use our simulations to infer CO emissions variability, identify bias in the CO emission inventory, and quantify the observed relative emissions (ORE) for the region. Finally, we identify the impacts of the COVID-19 shutdowns on CO emissions.

### 3.1 City-scaled Observed ΔCO

The observed ΔCO from the NYCMA varies substantially on sub-monthly timescales throughout the winter and spring and across all years of the study (Fig. 2). Mean 10-day observed ΔCO ranges from ~75 ppb to ~275 ppb, excluding the 2020 COVID-shutdown period (see Sect. 3.3). The winters of 2019, 2020, and 2022 experienced extended large peaks (>100 ppb) in observed ΔCO with general declines toward spring, while in 2021 the large peak occurs in late March (after COVID-19 related school closures ended), at the beginning of the transition to spring, and observed ΔCO increased again by May. In 2019
and 2020, there is less variability outside of these extended peaks as compared to 2021 and 2022, which show several additional small episodes of more elevated ΔCO (~50 ppb).

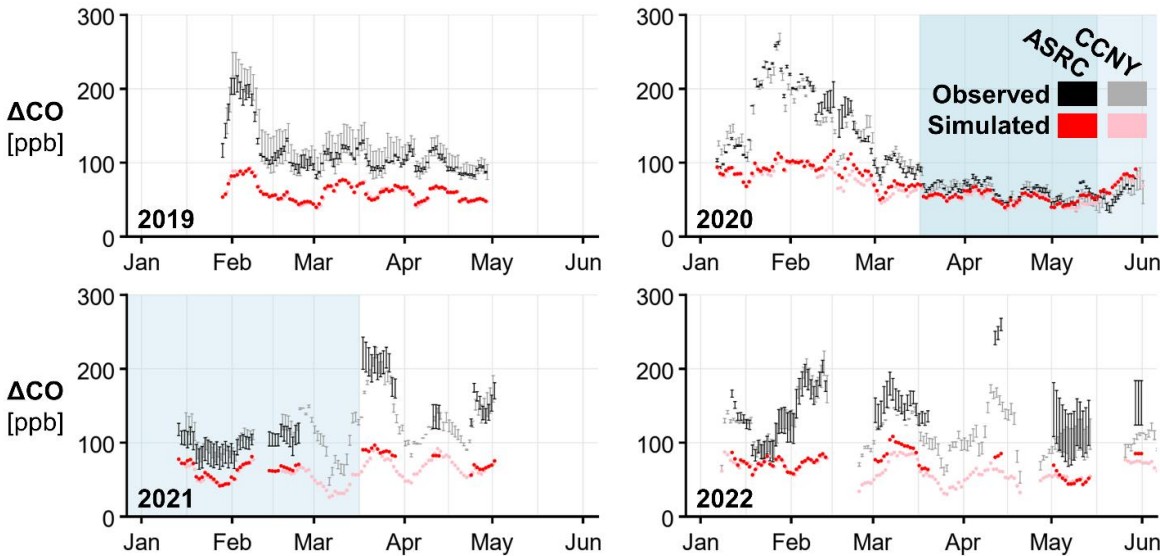

**Figure 2.** Timeseries of 10-day mean observed (black/grey) and simulated (red/pink) ΔCO for the New York City Metropolitan Area (NYCMA) domain at the ASRC (black/red) and CCNY (grey/pink) sites during January–May 2019–2022. The observed ΔCO vertical range
represents the uncertainty in the background CO. ΔCO are plotted at the center of the 10-day averaging period. The COVID-19 shutdown periods are shaded in blue: peak (darker, 15 March–15 May 2020) and residual (lighter, 16 May 2020–15 March 2021).



Observed ΔCO at the ASRC and CCNY sites are generally consistent and highly correlated, indicating observed ΔCO presented here is representative of city-scale ΔCO. The observed ΔCO at the two sites overlap within the uncertainty presented

by the background CO methods, except for slight deviations (~25 ppb) observed in February 2020 and March–April 2022.

There is a clear diurnal pattern in the observed ΔCO, which is largely consistent between the ASRC and CCNY sites (Figs. 3a, S5). Weekday (Monday–Friday) observed ΔCO peaks in the mid-to-late morning (6–12h EST), falls off in the early afternoon (12–15h EST), and peaks again in the evening (16–21h EST). Weekend (Saturday–Sunday) observed ΔCO tends to

have a lower peak (or no peak at all) in the late morning. The magnitude and amplitude of the diurnal pattern varies between year and season.

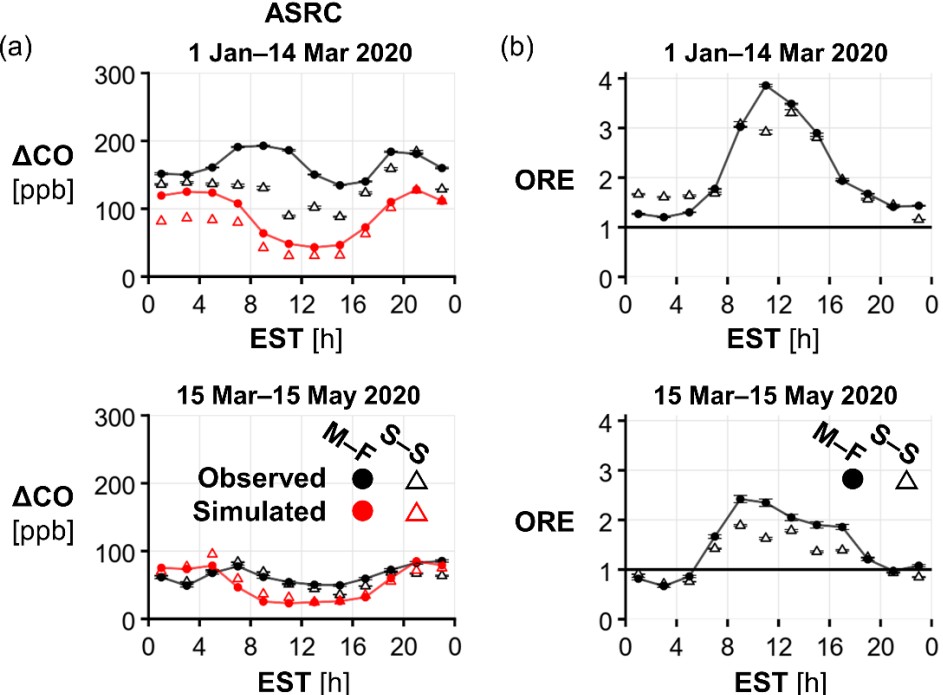

**Figure 3. (a)** Diurnal timeseries of mean observed (black) and simulated (red) ΔCO for the NYCMA domain at the ASRC site. ΔCO are separated by weekday (Monday–Friday, M–F; circles) and weekend (Saturday–Sunday, S–S; triangles) and then averaged every two hours
for the pre-shutdown 2020 (1 January–14 March 2020, top) and peak COVID-19 shutdown (15 March–15 May 2020, bottom) periods. Observed ΔCO is plotted as the mean using the two background CO methods, with the vertical bars representing their range. **(b)** Diurnal time series of observed relative emissions (ORE), where ORE is the ratio of observed ΔCO to simulated ΔCO using EDGAR from **(a)** for each period. ORE is plotted in the same matter as the observed ΔCO in **(a)**.

The uncertainty in the observed ΔCO derived from the different background CO calculation methods spans near 0 ppb to 50 ppb for the 10-day means, with most uncertainties ~10–25 ppb. This uncertainty varies between period and site. For example, the range of background CO is consistently larger for the CCNY site than the ASRC site in 2019, but the CCNY background





range is smaller than ASRC in 2021. The uncertainty is notably small at both sites during the maximum observed ΔCO of early winter 2020 and throughout the peak COVID-19 shutdown, while the largest uncertainty occurs during May 2022 at the

ASRC site. The uncertainty in observed ΔCO associated with the mean diurnal patterns is the same as noted for the 10-day means.

## 3.2 Inferred CO Emissions and Variability

Clearly there is more variability in the observed ΔCO than can be explained by the expected changes in the EDGAR CO emissions alone, which drop only 22% from February to May. To interpret the variability of the observed ΔCO and to evaluate

the inventory, we compare the observed ΔCO from the NYCMA with the simulated ΔCO, which combines emissions with transport meteorology (Fig. 2). The 10-day mean simulated ΔCO is always low (slope=~0.3) in magnitude compared to the observations and shows moderate variability (Figs. 4a–b), except for during and after the 2020 COVID-shutdown period (see Sect. 3.3). The simulated ΔCO spans only 50 to 100 ppb, resulting in a consistent underestimate of 50 ppb that can reach 150 ppb during times of peak observed ΔCO. This strong bias is present regardless of the observation site to which the simulated

ΔCO is sampled (or left unsampled) and despite the use of annual mean instead of monthly varying CO emissions (Fig. S6). The underestimation is also larger than the interannual variability of simulated ΔCO to changing meteorological conditions, which suggests the CO emissions from the inventory are the cause of the negative bias, rather than errors in the transport (Fig. S6). Results from sensitivity tests for January–February 2022, a period of highly variable observed ΔCO, further indicate that the uncertainty in transport does not account for the underestimate in simulated ΔCO (Fig. S7). We find that lowering the

minimum mixing height in STILT from 250 m to 150 m increases simulated ΔCO by only ~20 ppb and changes due to the alternative horizontal turbulence and vertical mixing schemes are near zero. All additional meteorological products we tested in STILT are generally consistent with the variability using HRRR but still do not reproduce the large peak in observed ΔCO in February 2022. Driving STILT with NAMS rather than HRRR increases simulated ΔCO by up to ~30 ppb during times of higher surface influence, and GFS and GDAS either match or are ~30 ppb lower than the simulated ΔCO driven by HRRR.


Excluding the COVID-19 shutdown periods, nearly 60% ($R^2$=0.52–0.59) of the variability in observed ΔCO is explained by the variability in simulated ΔCO, which is mostly driven by differential meteorology and transport to the ASRC site (Fig. 4a). The similar correlation ($R^2$=0.58–0.61) between the observed ΔCO and the relative NYC surface influence from HRRR-STILT, which is independent from CO emissions, explicitly confirms the impact of transport on observed ΔCO (Fig. 4c).

These results are consistent with the comparison at the nearby CCNY site (Figs. 4b, 4d). Excluding the 2020 peak and residual COVID-19 shutdowns, the distribution of relative NYC surface influence spans a large range that matches the full range of observed ΔCO at the ASRC and CCNY sites (Figs. 4c–d). This comparison indicates that air measured during times of higher observed ΔCO tends to experience more interaction with the NYC surface (and the subdomain's larger emissions). Therefore, the observed ΔCO for the NYCMA is heavily dependent on influence from the NYC subdomain, regardless of CO emissions



magnitude or variability. For the 10-day mean comparison, the variability in the simulated ΔCO induced by using monthly

varying rather than mean annual emissions is only 2.4% outside of the COVID-19 shutdowns (Fig. S8).

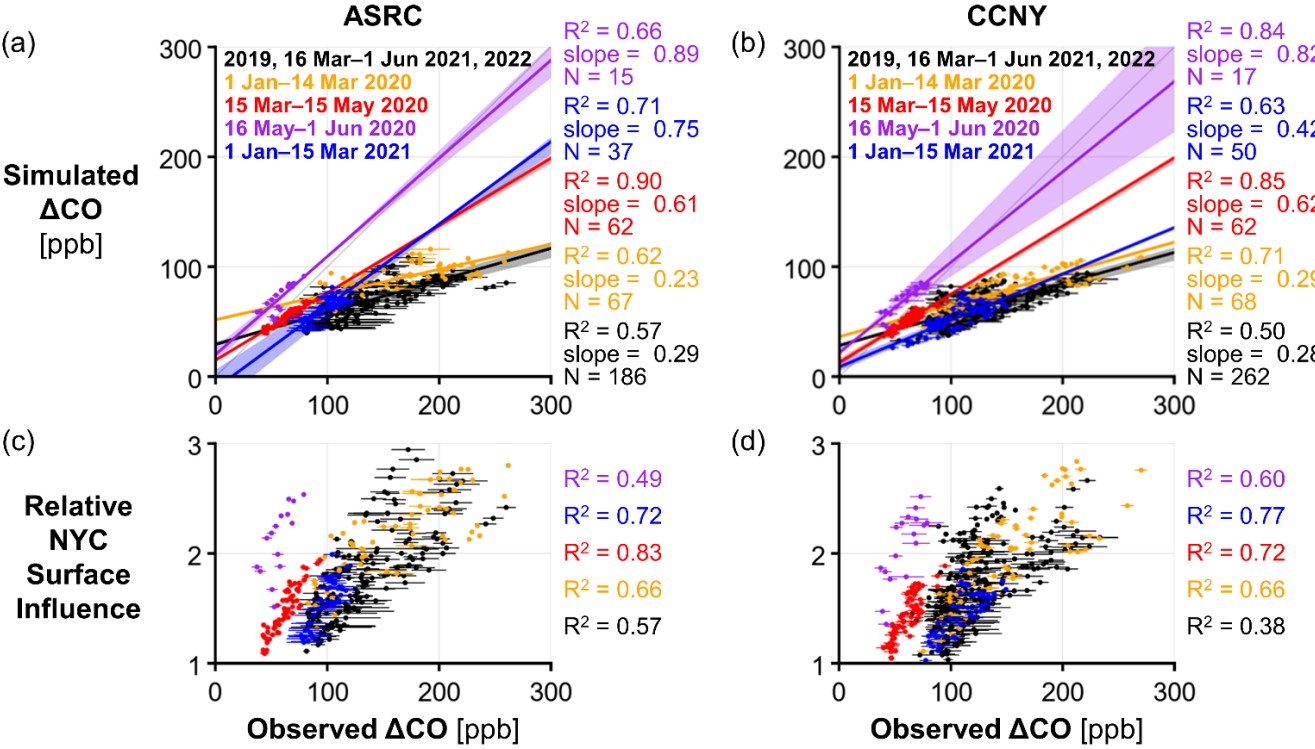

**Figure 4. (a)** Comparison of observed and simulated ΔCO at the ASRC site using ΔCO as in Fig. 2 for various periods surrounding and during the COVID-19 shutdowns: non-shutdown (2019, 16 March–1 June 2021, 2022; black), pre-shutdown 2020 (1 January–14 March
2020; yellow), peak shutdown (15 March–15 May 2020; red), residual shutdown 2020 (16 May–1 June 2020; purple), and residual shutdown 2021 (1 January–15 March 2021; blue). Observed ΔCO is plotted as the mean using the two background CO methods, with the horizontal bars representing their range. For each period, the linear best fit line and range from the background uncertainty, the slope determined by ordinary least squares, the coefficient of determination ($R^2$), and the number of points considered (N) are shown. The 1:1 line is shown in dark gray. **(b)** Same as in **(a)** but at the CCNY site. **(c)** Comparison of observed ΔCO at the ASRC site as in **(a)** and relative NYC surface
influence at the ASRC site for the periods as in **(a)**. For each period, the coefficient of determination ($R^2$) is shown. **(d)** Same as in **(c)** but at the CCNY site.

The remainder of the variability (~40%) in observed ΔCO is then attributed to actual changes in emissions on sub-monthly

timescales not included in the EDGAR CO emissions inventory. We can use the observed relative emissions (ORE), which is

the observed ΔCO normalized by the simulated ΔCO (see Sect. 2.9), to quantify these changes. ORE calculated using the 10-

day mean indicates the observed CO emissions are 1–3 times the magnitude in the EDGAR inventory, excluding the 2020

peak and residual COVID-19 shutdowns, and as with the observed and simulated ΔCO, the ORE is quite variable between

months and years (Fig. 5). In 2019, ORE is above 2 in winter and gradually declines to ~1.75 by spring. The winters of 2020

and 2022 experience short-lived peak ORE over 2.5, with a return to high ORE in spring 2022, although this period is more

uncertain given the range of background CO. The diurnal pattern of ORE shows a clear daytime peak and an even greater





range than using the 10-day means, with maximum CO emissions nearly 4 times EDGAR CO for weekdays (weekends are slightly lower) prior to the peak COVID-19 shutdown in 2020 (Fig. 3b). Substantial variability in CO emissions has been observed over short timescales due to variable sampling and is not unexpected given strong diurnal patterns of combustion sources (Lopez-Coto et al., 2020; Ren et al., 2018). The diurnal pattern of emissions observed here could be applied as city-specific scaling factors to CO emissions from all sectors (e.g., Crippa et al. (2020)).

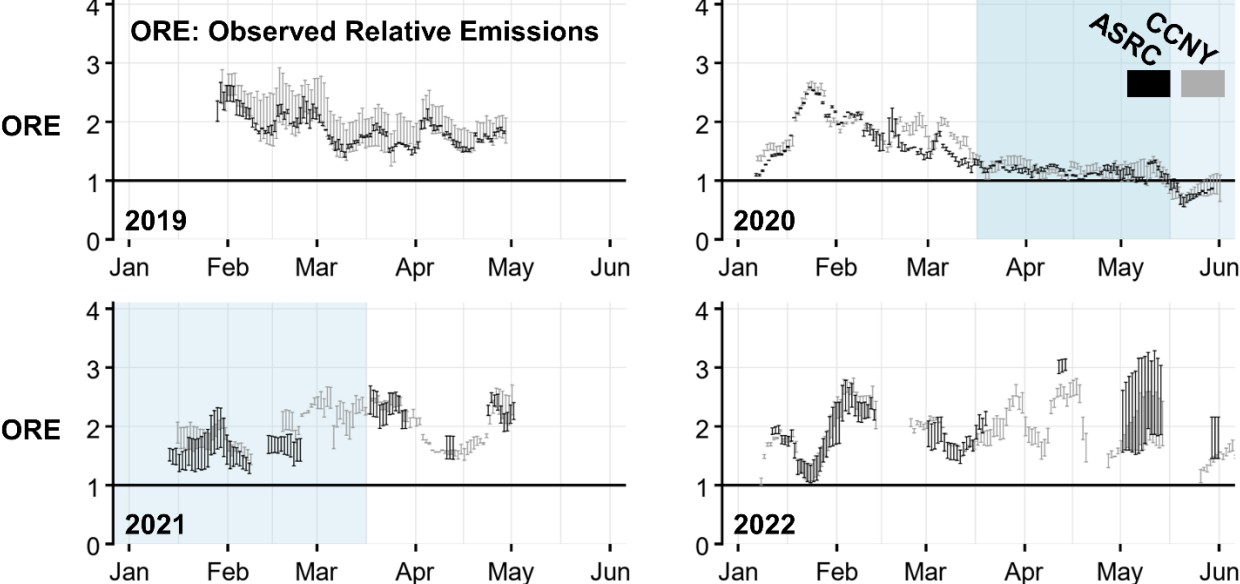

**Figure 5.** Timeseries of observed relative emissions (ORE) for the NYCMA domain at the ASRC (black) and CCNY (grey) sites during January–May 2019–2022, where ORE is the ratio of observed ΔCO to simulated ΔCO using EDGAR from Fig. 2. The vertical range represents the uncertainty in the background CO. ORE is plotted at the center of the 10-day averaging period. The COVID-19 shutdown periods are shaded in blue: peak (darker, 15 March–15 May 2020) and residual (lighter, 16 May 2020–15 March 2021).

On average, not including the COVID-19 shutdowns, the observed CO emissions for the NYCMA are 95% (81–109% varying the background CO) and 90% (78–102%) higher than in the EDGAR inventory at the ASRC and CCNY sites, respectively. These observed CO emissions estimates consistently lie substantially above the EDGAR uncertainty threshold of 44% and may be more in line with emissions from NEI, which were previously thought to be too high (e.g., Salmon et al. (2018)). There is no clear trend in observed CO emissions during our study period between the subsequent winters and springs not impacted by COVID-19 shutdowns, which is inconsistent with the US national trend of CO emissions reductions from transportation (e.g., Yin et al., (2015)) and suggests other combustion source sectors are a contributing cause of interannual variability.

### 3.3 COVID-19 Impacts on NYCMA CO

The lowest observed ΔCO (~50 ppb) during our study occurs during and after the peak COVID-19 shutdown in March–May 2020 (Fig. 2) and is partly driven by meteorology since the relative NYC surface influence during shutdown is on the lower





half of the overall distribution (Figs. 4c–d). Together, the low observed ΔCO and low relative NYC surface influence during the peak COVID-19 shutdown correspond to a regime that does not occur in any other years. The 2020 COVID-19 shutdowns are also the only period during our study when the magnitude of observed and simulated ΔCO nearly agree (ORE=~1, Fig. 5). 380   The large drop in observed CO emissions (38% at the ASRC site) during the peak COVID-19 shutdown compared to non-shutdown periods is consistent with changes in NO$_x$ emissions from Tzortziou et al. (2022). Matching the observed 60% reduction in mobility requires a doubling of the EDGAR CO on-road transportation emissions, in which case the modified inventory still underestimates the magnitude of the observed ΔCO peaks in winter.

A higher proportion of variability within the peak COVID-19 shutdown can be attributed to meteorology (85–90%) than other times, which implies relatively constant emissions (Figs. 4a–b). Only during the residual shutdown of May 2020 is the simulated ΔCO greater than the observed ΔCO (ORE<1), after the loosening of some restrictions (Figs. 2, 5). Here the CO emissions remain lower than expected by EDGAR, despite the relative rise in observed and simulated ΔCO due to a turn toward greater sampling of the NYC surface (Figs. 4c–d). This result may be due to the transition away from heating as 390   temperatures increased in May 2020.

We also observe lower CO emissions compared to other years during the less severe residual COVID-19 shutdown in winter 2021, when observed CO emissions are only 59% greater than the EDGAR inventory at the ASRC site, a 16% drop from non-shutdown periods. This result is consistent with lingering restrictions on mobility and reduced observed NO$_2$ (-30% compared 395   to 2018–2019). The larger relative reduction of NO$_2$ compared to the expected emissions changes is due to favorable wind conditions, according to Tzortziou et al. (2022). However, the distribution of relative NYC surface influence from our study during the residual shutdown of winter 2021 is not particularly different from during the peak shutdown (Figs. 4c–d). This example highlights the importance yet difficulty of considering meteorology when connecting emissions changes to atmospheric measurements, especially during periods of rapid change.

There is a clear reduction in magnitude and distinct flattening of the observed and simulated ΔCO diurnal patterns during the COVID-19 shutdowns compared to non-shutdown periods (Figs. 3a, S5). Much of this magnitude change is due to transport differences, and so the ORE is particularly useful here to see the relative change in emissions (Fig. 3b). The peak COVID-19 shutdown leads to a decrease of ~35% in maximum weekday daytime CO emissions (ORE drops from 4 to 2.5), likely due to 405   reduced traffic emissions, and extends elevated emissions compared to the overnight minimum an hour earlier and later, indicating a longer rush hour in the city. The weekend daytime maximum emissions reduction during the peak COVID-19 shutdown is even greater, nearly 50%. Only during the overnight hours during this peak shutdown do the observed CO emissions consistent with the EDGAR CO emissions magnitude.



## 4 Conclusions

Continuous in-situ observations are useful to quantify and characterize highly variable ΔCO from urban domains such as the NYCMA, especially during emission-source transition periods between winter and spring. This observed ΔCO variability is heavily dependent on meteorology and transport, and these factors must be accounted for in air quality studies that try to connect atmospheric observations with emissions changes. In the NYCMA, we found a substantial portion of observed ΔCO changes caused by emissions variability after removing this weather dependence.


Multiple years of observations capture different seasonal and diurnal patterns in inferred CO emissions that are generally not accounted for in CO emission inventories. More variability is needed in these inventories, which could reasonably be developed and implemented for diurnal patterns using city-specific observations. However, large variability in day-to-day vehicle emissions seems unlikely outside of weekday/weekend effects and extraordinary events such as the COVID-19 pandemic.

Therefore, at least in the NYCMA, the combustion source variability on the seasonal and sub-monthly scale lies elsewhere, from sectors such as building heating/cooling and electricity generation, although the latter is tightly monitored for inefficiencies at large power plants.

The EDGAR CO emissions evaluated here for the NYCMA are greatly underestimated. Studies that scale greenhouse gas
emissions by combining observed $CO_2$:CO or $CH_4$:CO with CO inventories could also be underestimating those emissions by 2–3 times. We encourage evaluating the emissions inventory magnitude itself, rather than only the ratios, for the specific study period and location, since the emissions ratios and their trends are uncertain and can vary between inventory year and version.

The COVID-19 shutdowns removed substantial vehicle traffic from the NYCMA, and only then did the inventory match the
CO emissions inferred from observations. When the observed drop in transportation during the peak shutdown is reproduced in the inventory, the required increase in CO emissions is still not enough to match the observed ΔCO peaks, which shows that other non-transportation emissions (e.g., stationary sources) are larger than currently accounted for in this inventory. These sources must be addressed in the NYCMA if there is any hope of reducing carbon fuel combustion emissions for air quality improvement (CO) and to meet greenhouse gas emission targets (co-emitted $CO_2$ and $CH_4$).

**Data availability**

Data that support the findings of this study are available as listed below:

ASRC Rooftop CO observations, NYCMA observed and simulated ΔCO, and relative NYC surface influence: https://datadryad.org/stash/share/d8sHZ76SE5JcNHozuCYLt0vquwkfhbea5WVKFURu9Aw

EPA CO observations: https://aqs.epa.gov/aqsweb/airdata/download_files.html

EDGAR CO emissions: https://edgar.jrc.ec.europa.eu/index.php/dataset_ap61



STILT model: https://uataq.github.io/stilt/#/

HRRR ARL files: https://www.ready.noaa.gov/archives.php

**Author contributions**

LDS and RC designed the study. RC and AHD provided calibrated ASRC CO measurements. CC and RTC supported ASRC
CO measurements. LDS performed STILT simulations and primary analysis of observation-model comparison. RC and BD
assisted the analysis. LDS wrote the paper. All co-authors contributed to the preparation of the manuscript.

**Competing interests**

Authors declare that they have no competing interests.

**Acknowledgements**

This work was funded by NOAA research grants (LDS, RC, CC: NA20OAR4310306; LDS, RC: NA21OAR4310235). LDS
and RC were also funded by support from the Columbia University Department of Earth and Environmental Sciences. The
authors thank J. Mak for instrumentation and the Advanced Science Research Center (ASRC) for hosting the measurements
at the ASRC Rooftop Observatory. We also thank the STILT development team and the R Project community for analysis and
plotting tools, especially the ggplot2, ggpattern, magick, anytime, lubridate, raster, and cowplot packages.

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
