# Peer review of "Multi-year observations of variable incomplete combustion in the New York megacity"

_EGUsphere, 2024_

## Author Comment (AC1)

**Author Response to RC1**

General comments:

Schiferl et al. present an analysis of urban CO enhancements by using in-situ data from multiple sites collected over several years. The study also relies on HRRR-STILT atmospheric modelling and the global EDGAR CO emission inventory. Key results presented are that, a.) CO emissions in the NYCMA are underestimated, b.) there was a significant impact on CO emissions during COVID shutdowns, c.) that stationary combustions are also likely underestimated.

Overall, the paper is well-written, although key results could be emphasized more. The data and topic is of interest to the atmospheric communities as emissions of air pollutant and (indirect) greenhouse gases need to be better understood to allow interventions for e.g. air quality control measures. There are some significant issues that should be addressed before publication, but after those have been addressed the paper is surely suitable for publication in ACP and will be of interest to its readership.

We thank the reviewer for their helpful comments and suggestions. Specific responses to each comment follow in red, with edits to the manuscript in blue. Line numbers refer to the original manuscript.

Major comments:

The EDGAR inventory is a global data product and not intended for use on an urban scale. Given it limited suitability of EDGAR at this scale needs more discussion, especially of the limitation.

We acknowledge that the global and national-scale assumptions made in EDGAR may not be ideal for urban regions such as the NYCMA and have added this acknowledgement to the manuscript. However, one main purpose of this study is to evaluate the CO inventory previously used for greenhouse gas scaling over multiple cities in the US (e.g., Plant et al. 2019). EDGAR was used in our study and in those previous studies because it is the only inventory with both CO and greenhouse gas emissions within a consistent framework. We considered other inventories such as US EPA NEI (documented high bias) and FIVE (mobile sources only, see response to Reviewer 2), but due to their noted weaknesses and lack of consistent greenhouse gas emissions information, we have not included them. These considerations are also described in the introduction at lines 58 and line 70.

Plant, G., Kort, E. A., Floerchinger, C., Gvakharia, A., Vimont, I., and Sweeney, C.: Large Fugitive Methane Emissions From Urban Centers Along the U.S. East Coast, Geophys. Res. Lett., 46, 8500–8507, https://doi.org/10.1029/2019GL082635, 2019.

Added to Sect 2.6: "While not ideal for urban-scale studies such as in the NYCMA, EDGAR provides the best option for linking CO and greenhouse gas emissions."

Given that multiple instruments are used across the network and at the same sites is it important to better describe the quality control and merging of data needs to be more transparent. How well do the different instruments compare and how stable are the measurements provided by the lower resolution instruments?

The EPA AQS sites included in this analysis use (i) a Teledyne API 300EU analyzer at CCNY, NYC and (ii) a 48i-TLE enhanced trace level CO analyzer in Cornwall, CT that both report hourly CO dry-mole fractions to 1 ppbv precision. The calibrated ASRC site CO observations are averaged to match the resolution of the calibrated AQ (CCNY and Cornwall) instruments: hourly temporal, 1 ppbv precision. We will clarify the precision of the reported values from the instruments in the text.

The NIST and NOAA/WMO scales differ by up to 5 ppb (line 123), but the difference in scales does not change the results of our ASRC-CCNY inter-site comparison (i.e., the observed CO and ΔCO at both sites is much greater and more variable than 5 ppb). We note that at Cornwall the analyzer is self-calibrated to the NIST standard daily (line 121) while the ASRC instruments (Picarro G2401 and Aerodyne SuperDUAL) were calibrated to the NOAA/WMO scale (line 109).

For merging the datasets, previous studies have shown that the CO mole fraction reported by instruments of various precision can be used interchangeably once the instruments are calibrated to the same scale and the precision is incorporated (e.g., Thompson et al., 2022, Deeter et al., 2022, Hegarty et al., 2022). For this analysis, we did not have multiple instruments operating at the same time at any site but all instruments used at the ASRC observatory were calibrated to the NOAA/WMO scale in the same way, which ensured all data were processed consistently across years. The Aerodyne SuperDUAL has slightly higher precision for CO (< 0.05 ppbv at 60s, Commane et al., 2022) compared to the Picarro G2401 (<0.4 ppbv at 5 min), but the consistent calibration of the instruments ensures no bias was introduced.

Thompson et al., The NASA Atmospheric Tomography (ATom) Mission: Imaging the Chemistry of the Global Atmosphere, BAMS, 103(3), E761-E790. https://doi.org/10.1175/BAMS-D-20-0315.1, 2022.
Deeter, M., Francis, G., Gille, J., Mao, D., Martínez-Alonso, S., Worden, H., Ziskin, D., Drummond, J., Commane, R., Diskin, G., and McKain, K.: The MOPITT Version 9 CO product: sampling enhancements and validation, Atmos. Meas. Tech., 15, 2325–2344, https://doi.org/10.5194/amt-15-2325-2022, 2022.
Hegarty, J. D., Cady-Pereira, K. E., Payne, V. H., Kulawik, S. S., Worden, J. R., Kantchev, V., Worden, H. M., McKain, K., Pittman, J. V., Commane, R., Daube Jr., B. C., and Kort, E. A.: Validation and error estimation of AIRS MUSES CO profiles with HIPPO, ATom, and NOAA GML aircraft observations, Atmos. Meas. Tech., 15, 205–223, https://doi.org/10.5194/amt-15-205-2022, 2022.
Commane, R., Hallward-Driemeier, A., and Murray, L. T.: Intercomparison of commercial analyzers for atmospheric ethane and methane observations, Atmos. Meas. Tech., 16, 1431–1441, https://doi.org/10.5194/amt-16-1431-2023, 2023.

Added in Sect. 2.1: "…which are rounded to the nearest 1 ppbv."
Added in Sect. 2.2: "Both sites report CO dry-mole fractions at 1 ppbv precision."

Minor and technical comments:

Line 26: What is the threshold for an exceedance event?

The text has been modified to include the hourly exceedance threshold of 35 ppm, which is appropriate for our analysis of hourly CO observations.

The text starting in line 25 now reads: "Poisonous to humans at high concentrations, CO is a regulated criteria pollutant in many countries including in the United States (US), where the exceedance event threshold is an hourly average of 35 ppm."

Line 49: What is the underlying process suggested here for this positive correlation with temperature and humidity, is this only for moving traffic or also cold starts emissions?

The observed relationship is likely due to the temperature sensitivity of pollution control equipment on diesel vehicles. We clarify that this relationship is relative to $NO_x$ emissions, which decrease with increasing temperature.

Line 49 now reads: "Hall et al. (2020) found evidence for a vehicle emissions dependence on meteorology, where CO emissions relative to nitrogen oxides ($NO_x$) increased with increasing temperature across the winter-to-summer gradient in the DC/Baltimore region, likely due to the temperature sensitivity of pollution control equipment on diesel vehicles."

Line 79: Would seem reasonable to include Monteiro et al. multi-city study here? (DOI 10.1088/2515-7620/ac66cb)

Yes, the citation for Monteiro et al. has been added.

The text starting in line 77 now reads: "These reductions were noted on the global scale for CO2 (Forster et al., 2020; Le Quéré et al., 2020; Liu et al., 2020) and on the city scale for NO2 (Goldberg et al., 2020; Shi et al., 2021; Tzortziou et al., 2022), CO (Lopez-Coto et al., 2022; Monteiro et al., 2022), and CO2 (Monteiro et al., 2022; Turner et al., 2020)"

Monteiro, V., Miles, N. L., Richardson, S. J., Turnbull, J., Karion, A., Kim, J., Mitchell, L., Lin, J. C., Sargent, M., Wofsy, S., Vogel, F., and Davis, K. J.: The impact of the COVID-19 lockdown on greenhouse gases: a multi-city analysis of in situ atmospheric observations, Environ. Res. Commun., 4, 041004, https://doi.org/10.1088/2515-7620/ac66cb, 2022.

Line 107: Can we see the timeseries to be sure no bias was introduced by using different instruments? how about changing measurement frequencies?

The hourly time series of the ASRC CO observations is shown in Fig. 1. Regardless of the measurement frequency of each analyzer, all data are averaged to an hourly basis to match the reported times of the CCNY data (line 114).

No changes have been made to the text.

Line 121: According to its specification sheet the 48i-TLE is not comparable to the Picarro instruments it has a 24h drift of <100ppb and a span drift of 1% full-scale <100000ppb. See also general comments.

See response to related major comment above.

Line 123: The offset between WMO and NIST scale is indeed small, but that does not deal with the limited resolution and potential drift of the AQ instruments.

See response to related major comment above. We find largely the same results for both New York City sites (e.g., Fig. 2), even though the site types have different native precisions.

Added in Sect. 2.1: "…which are rounded to the nearest 1 ppbv."

Added in Sect. 2.2: "Both sites report CO dry-mole fractions at 1 ppbv precision."

Line 132; Why was a threshold of 3 sigma chosen here? Why not 2 sigma or 2.5 sigma?

We chose 3 sigma since it is the typical threshold used to define statistical outliers beyond 99.7% of points in the distribution. Three sigma is a more conservative choice than 2 or 2.5 sigma, meaning more data is retained in the analysis.

No changes have been made to the text.

Figure 1; What explains the very strong increase in CO variability at the ASRC site in 2021 and 2022?

The increase in CO variability at the ASRC site in 2021 and 2022 is due to greater intensity and frequency of local-scale plumes observed from nearby buildings. Our city-scale filter approach described in Sect. 2.3 removes these local-scale plumes from the analysis.

No changes have been made to the text.

Line 170: Earlier you report the instruments provide measurements in ppbv, now calculations are done in ppb. Did you correct for the fact that air is not an ideal gas? Only for mixtures of ideal gases are ppbv and ppb (i.e part per billion per mole) equivalent.

The correct units are pbbv.

The units have been changed where needed throughout the text and figures.

Line 180: Why EDGAR emissions not compared to high-resolution CO data from FIVE for NYC? NYICE 2018 WRF-Chem Emission Data (noaa.gov)

The Fuel-based Inventory of Vehicle Emissions (FIVE) provides estimates of the on- and off-road components of transportation emissions using fuel usage data. FIVE is available at a higher spatial resolution (4km) than the EDGAR emissions used in our study but is presented at the same monthly temporal resolution. Emissions scaling to fuel usage in FIVE is done at the state-, rather than city-level, and so FIVE may not provide true greater benefits at higher spatial resolution when considering urban-scale changes.

An updated version of FIVE (more recent than the NYICE 2018 WRF-Chem Emissions Data cited by the reviewer) is available from https://csl.noaa.gov/groups/csl7/measurements/2020covid-aqs/emissions/ and described by Harkins et al. 2021 for 2019-2021 of our study period, including both for both business-as-usual and COVID-19 impact scenarios for 2020. Results comparing observed $\Delta CO$ to simulated $\Delta CO$ using FIVE for the road transportation sector CO emissions rather than EDGAR are shown in Figure R1. FIVE on-road emissions improve the comparison slightly (~20% increase in magnitude) compared to using only EDGAR for all sectors, but there is no improvement in the temporal comparison – wintertime peaks remain underestimated. Additionally, we want to focus our evaluation on EDGAR, which has been used for greenhouse gas scaling – FIVE is only for one segment (transportation) and not for greenhouse gases. For these reasons, we choose not to include FIVE in our paper.

[Figure]

Figure R1. Time series of 10-day mean observed ΔCO (black/grey) and simulated ΔCO using total EDGAR CO emissions (red/pink) for the NYCMA domain as in Fig. 2. Additional simulated ΔCO versions are shown: (i) EDGAR CO emissions without road transportation section (non-road, orange shading), (ii) FIVE on-road transportation with EDGAR non-road CO emissions (blue/cyan), and (iii) the change in FIVE on-road transportation CO emissions induced by the COVID-19 shutdown in 2020 (purple/magenta).

Harkins, C., B. C. McDonald, D. K. Henze, and C. Wiedinmyer, A fuel-based method for updating mobile source emissions during the COVID-19 pandemic, Environ. Res. Lett., doi:10.1088/1748-9326/ac0660, 2021.

No changes have been made to the text.

Line 204: How do you know that biomass emissions in winter are negligible for NYCMA, especially in the suburbs?

Limited biomass burning emissions (i.e., from prescribed burning and wildfire) were calculated from the GFED4s biomass burning emissions inventory as cited in the text at line 205. If the reviewer is referring to biofuel emissions (i.e., from burning of wood for home heating or energy), these emissions are already included as anthropogenic sources from fuel combustion in the EDGAR CO emission inventory. We will amend the text to clarify that line 204 refers to prescribed fire and wildfire emissions from biomass burning.

Line 204 now reads: "CO emissions from biomass burning (i.e., prescribed burning and wildfire) sources are very limited during the winter and spring in the NYCMA, and we do not consider them here."

Line 244: unnecessary "(" before NAMS or missing ")" in line 246

Fixed.

Added missing ")" in line 246.

Line 398: Isn't part of the inability to properly quantify meteorological influence due to the fact that data has been heavily pre-processed? 10 day averages seem to by definition limit the ability to understand rapid meteorological changes?

Averaging periods certainly may play a role in the mismatch referred to here. Tzortziou et al., 2022 averaged the data into even courser time segments (monthly compared to 10-day bins in our study). As "rapid" is a relative term and for clarification, we have removed the clause "especially during periods of rapid change" and instead rephrased line 398 to emphasize the averaging mismatch as a potential reason for the differential outcomes between our two studies.

Tzortziou, M., Kwong, C. F., Goldberg, D., Schiferl, L., Commane, R., Abuhassan, N., Szykman, J. J., and Valin, L. C.: Declines and peaks in $NO_2$ pollution during the multiple waves of the COVID-19 pandemic in the New York metropolitan area, Atmos. Chem. Phys., 22, 2399–2417, https://doi.org/10.5194/acp-22-2399-2022, 2022.

In line 395, we clarified that the $NO_2$ reductions in Tzortziou et al., 2022 were "calculated on a monthly basis". The text starting in line 397 now reads: "The contrasting outcomes from these studies highlight the importance of considering meteorology and time averaging when connecting emissions changes to atmospheric measurements."

Line 433: There is no need to improve CO inventories to reduce GHG emissions. Policies like electric vehicles, better public transit, nuclear, solar or wind power can be implemented without any knowledge of CO emissions.

The reviewer is correct that CO inventories do not need to be improved to reduce GHG emissions. However, line 433 refers to addressing (i.e., identifying and mitigating) the unknown sources (possibly stationary sources) that emit both CO and GHGs in the NYCMA that are suggested by our analysis - not fixing the inventories themselves. We have modified the sentence to improve clarity.

The text starting in line 432 now reads: "The sources of these unattributed CO emissions must be identified in the NYCMA in order to mitigate carbon fuel combustion for both air quality improvement and, since $CO_2$ and $CH_4$ may be co-emitted with CO, to meet greenhouse gas emission targets."

**Author Response to RC1**

The authors have conducted an analysis using CO observations in New York City to characterize urban-scale variability in CO and to evaluate the EDGAR CO emission inventory. In particular, they take advantage of the close proximity of the observations located at the City College of New York campus and the observations at the Advanced Science Research Center to isolate the urban-scale variations from variations associated with local emissions. This is an interesting study that makes clever us of a limited number of observational sites to characterize variations in CO across the New York City metropolitan area. The paper is well written and the analysis is sound. I recommend publication of the manuscript after the authors have addressed my minor comments below.

We thank the reviewer for their helpful questions and comments. Specific responses to each comment follow in red, with edits to the manuscript in blue. Line numbers refer to the original manuscript.

Minor comments:
1. To assess the impact of meteorology on the analysis, the authors compared the HRRR-based results to those based on NAMS, GFS, and GDAS. But the latter three meteorology products were all available at coarse resolution (12 km for NAMS, 0.25 deg for GFS, and 1 deg for GDAS) so it is unclear to me how useful they are for the urban-scale analysis presented. Is it because of the resolution that we see larger differences for GFS and GDAS in Figure S7?

We performed the analysis in Fig. S7 using additional meteorology products coupled with STILT to assess the sensitivity of the study results to the choice of meteorological product. We found that the underestimate in simulated $\Delta$CO is unlikely to be caused be errors in the meteorological products, which reinforces the conclusion that there is a large underestimate in CO emissions from EDGAR (line 318). It is likely that the spatial resolution difference accounts for variability in simulated $\Delta$CO for STILT driven by GFS and GDAS, which may not be able to capture the land-sea gradients across the NYCMA.

No changes have been made to the text.

2. To remove the variability on synoptic time scales, the authors averaged the data with a 10-day window. However, averaging over 10 days could remove a lot of the transported related variability. How sensitive are the results to this 10-day averaging?

We tested a range of averaging lengths from 4 to 16 days and found that the results are relatively insensitive to this choice. $R^2$ between observed and simulated $\Delta$CO for non-shutdown periods ranges from 0.52 to 0.68 at the ASRC site and from 0.55 to 0.68 at the CUNY site with no consistent trend the length-choice gradient. The slope of the comparison ranges from 0.28 to 0.35 at the ASRC site and from 0.31 to 0.35 at the CUNY site.

Added in Sect 2.5: "For (i), we tested a range of 4-day to 16-day means and found little sensitivity in the results to the length of averaging period."

While investigating the sensitivity of our results to the 10-day assumption for averaging, we resolved an issue in the background calculation which resulted in the exclusion from analysis of much of the 2022 data. To preserve as much data as possible, we revised our method to include all data points with at least one available background, rather than requiring both. This change has minimal impact on the overall results.

Starting at line 170 now reads: "Observed $\Delta CO$ is calculated for each of the ASRC and CCNY sites using the observed CO from those sites and both the fifth percentile background for that site and the Cornwall background. [no changes, copied for context] To quantify the background uncertainty, we compare the calculated observed $\Delta CO$ for hours with valid background CO from both locations." Figure S3 has been updated to show the correct background CO mole fractions. Changes to show points with only one background were also made to Figures 2, 3, 4, 5, S5, S6, S7, and (now) S9.

3.  Lines 114 and 147-148: Why was a criterion of 50% valid data selected?

Similarly to the averaging lengths, we tested a range of valid data percentage thresholds from 30% to 70% and found that the results are relatively insensitive to this choice. $R^2$ between observed and simulated $\Delta CO$ for non-shutdown periods ranges from 0.42 to 0.68 at the ASRC site and from 0.55 to 0.56 at the CCNY site with no consistent trend the threshold-choice gradient. The slope of the comparison ranges from 0.24 to 0.31 at the ASRC site and from 0.30 to 0.31 at the CCNY site.

Added in Sect 2.4: "We tested a range of thresholds from 30% to 70% and found little sensitivity in the results to the minimum data availability percentage."

4.  Figure 3 is somewhat confusing and not well explained. Since the EDGAR inventory lacks sub-monthly variability, can we assume that the simulated diurnal variations on the CO enhancement and the ORE are due to transport variations? For example, the simulated enhancement has a minimum in the middle of the day. Is this due to diurnal variations in the PBL depth? If that is the case, the ORE in 3b reflects the influence of discrepancies in the EDGAR emissions as well as errors in the meteorology, in contrast to the statement on line 254 that the ORE "removes the impact of meteorology."

Yes, the reviewer is correct that "the simulated diurnal variations on the CO enhancement" are due to transport variations. And yes, those variations are likely largely impacted by changes in the PBL depth. ORE in this case assumes no "errors" in meteorology (and we have shown that the meteorological uncertainty is small compared to the underestimate in emissions, see Figs. S7 and S8). Therefore, diurnal variations in the ORE are due to the changes in the emissions that would be needed to create the difference between the observed and simulated enhancements. We will clarify that rather than removing the impact of meteorology, we are removing the *variability* of the meteorology to highlight the sensitivity of the simulated $\Delta CO$ to emissions.

Line 254 now reads: "…which removes the variability of meteorology and highlights the sensitivity of simulated $\Delta CO$ to the observed change in CO emissions for any period:"

5.  Figure S7 shows the variations in the 10-day mean enhancements for changes in the mixing heights. It would be helpful to see what is the impact of the mixing heights on the results shown in Figure 3. In addition, how do the diurnal variations in Figure 3 change when using the NAMS fields instead of HRRR?

Figure S8 (newly created) now shows the sensitivity of diurnal variations in Fig. 3 to choice of minimum mixing height (250 m v. 150 m) and meteorological product (HRRR v. NAMS). As with the 10-day mean comparison in Fig. S7, the underestimate in simulated ΔCO throughout the day is little affected by these alternative choices for STILT and does not make up for the observational discrepancy, especially during daytime. Lowering the minimum mixing height causes a small but consistent reduction in the ORE since more emissions are kept near the surface. NAMS causes a lower peak ORE that is shifted later in the day than HRRR, likely due to difficulties in capturing the timing of the diurnal mixed layer at the urban scale along a coastal gradient as in the NYCMA.

Added Fig. S8.

[Figure]

**Figure S8.** Diurnal timeseries of mean observed (black) and simulated (red) ΔCO (left) and observed relative emissions (ORE, black, right) for the NYCMA domain at the ASRC site as in Fig. 3 during January–February 2022. Additional simulated ΔCO from sensitivity tests of STILT shown in blue: reducing minimum mixing height from 250 m to 150 m (top) and using the NAMS (North American Mesoscale Forecast System) meteorological product to drive STILT instead of HRRR (High-Resolution Rapid Refresh) (bottom). Sensitivity of ORE to these tests shown in blue on right for both the minimum mixing height (top) and use of NAMS meteorology (bottom).

Added in Sect. 3.2: "Simulated ΔCO is similarly insensitive to minimum mixing height and meteorological product choice (only NAMS shown) on diurnal timescales (Fig. S8)."

Added in Sect. 3.3: "The diurnal pattern in ORE is sensitive to the choice of meteorological product, which is consistent with difficulties in capturing the timing of the diurnal mixing layer in coastal locations (Fig. S8)."

6. Line 407: should "do to the observed CO emissions" be changed to "are the inferred CO emissions"?

Yes, thank you.

Changed to "are the inferred CO emissions".